



# A survey of snow growth signatures from tropics to Antarctica using triple-frequency radar observations

Qinghui Li[1], Haoran Li[2], Xuejin Sun[1], Yun Zhang[1], Weitao Lyu[2,3], Zheng Ruan[2], Liping Liu[2], Aiming Liu[4], and Chunsheng Zhang[4]

[1]College of Meteorology and Oceanography, National University of Defense Technology, Changsha, China
[2]State Key Laboratory of Severe Weather Meteorological Science and Technology, Chinese Academy of Meteorological Sciences, Beijing, China
[3]CMA Key Laboratory of Lightning, Chinese Academy of Meteorological Sciences, Beijing, China
[4]Shenzhen Meteorological Observatory, Shenzhen, China

**Correspondence:** Haoran Li (lihr@cma.gov.cn)

**Abstract.** Snow formation is a complex interplay of multiple microphysical growth processes, and the prevailing snow characteristics are inherently linked to local climate. However, the persistent shortage of observations for characterizing snow microphysics at a global scale continues to constrain our understanding of snow growth processes. Here, we investigate snow riming and aggregation signatures in stratiform precipitation through triple-frequency radar observations collected during co-ordinated field campaigns across Southern China, the Eastern United States, Western Europe, Northern Europe and Antarctica. The results suggest that the velocity-based riming estimates are generally consistent with triple-frequency observations, and the riming frequency increases with temperature. Our analysis of dual-frequency observations in these field campaigns qualitatively indicate the dendritic growth zone around -15 °C playing a key role in initiating enhanced snow size growth, and reveals a generally temperature-dependent snowflake growth characteristics. The snow over Eastern US is characterized by the most prominent riming growth, corresponding to moderate to heavy riming. Triple-frequency signatures of snowflakes over west Europe are consistent with Southern China, while the latter shows a higher degree of riming. The weakest snow growth signatures were found over west Antarctica, potentially owing to the scarcity of ice nucleating particles and available water vapor for deposition. In addition, our statistics reveal a latitudinal dependence for snowfall detection limitations with current spaceborne Ku- and Ka-band radars, and shed novel insights into future triple-frequency satellite missions as well as joint application of weather and spaceborne radars.

## 1 Introduction

Half of the Earth's surface precipitation events originate from snow (Field and Heymsfield, 2015; Mülmenstädt et al., 2015; Heymsfield et al., 2020). Formation and growth of snow are governed by multiple interacting microphysical processes, such as, nucleation, secondary ice production, deposition, riming, aggregation, which take effect on various pathways depending on atmospheric dynamics and thermodynamics (Gultepe et al., 2000). Hence, the prevailing snow characteristics (size, shape, density, falling velocity, etc) are dependent on regional atmospheric conditions, such as temperature, humidity, air motions, etc.



For instance, aggregation is significantly enhanced around the melting layer (Dias Neto et al., 2019) and is affected by vertical air motions (Von Terzi et al., 2022); riming is favorable in shallow Arctic mixed-phase clouds at temperatures greater than -12 °C (Morrison et al., 2012; Verlinde et al., 2013; Kneifel and Moisseev, 2020; Fitch and Garrett, 2022; Maherndl et al., 2023) and is necessary for graupel formation (Li et al., 2018; Zhang et al., 2021). The concentration of ice nucleating particles (INPs) largely depends on temperature, in addition to other factors such as aerosol compositions and concentrations, and presents significant site-to-site variations (DeMott et al., 2010; Kanji et al., 2017).

In spite of the importance of regional climate to snow microphysics, explicitly representing microphysical processes in models is challenging owing to our knowledge gaps in cloud physics as well as the simplified microphysics schemes that are inherently uncertain and lack observational constraints (Morrison et al., 2020; Liu et al., 2023). For example, the parameterized aggregation efficiency in many microphysics schemes monotonically increase with temperature (Zhang et al., 2024), while an increased sticking efficiency at around -15 °C is more consistent with actual observations (Karrer et al., 2021). Although riming has been implemented in some microphysics schemes (Lin et al., 2011; Morrison and Milbrandt, 2015), the interactions between riming and aggregation as evidenced in recent observations (Li et al., 2020; Chellini et al., 2022) suggest the need of more delicate tunning of snow microphysics in models. Therefore, adequate observations are required for disentangling the process-level understanding of snow microphysics and improving the representation of specific physical processes in numerical models.

Major responsible microphysical processes for snow characteristics can be inferred from in-situ observations. In well-designed field campaigns, ice particles in natural clouds have been extensively recorded with in-situ probes. Since these instruments are mounted on specific platforms, such as balloons, aircrafts, or cable cars, profiling the microphysical processes taking place in clouds is complemented by the means of remote sensing. Thanks to the development of active remote sensing techniques, meteorological radars have shown promise in characterizing snow microphysics. Since the dawn of meteorological radars, snow has been identified from radar echoes, and then radars have proven to be a unique tool for observing snow (Maynard, 1945; Hooper and Kippax, 1950). In recent decades, significant improvement has been made in quantitative characterization of snow microphysics thanks to the advances in understanding the scattering characteristics of snow as well as the implementation of Doppler, dual-polarization and multi-frequency radar techniques (Bringi and Chandrasekar, 2001; Kneifel et al., 2015; Mason et al., 2018; Tridon et al., 2019, among others).

The basis of radar remote sensing of snow microphysics lies in the interactions between electromagnetic waves and ice particles, known as scattering. If the maximum dimension (D) of a particle is much smaller than the radar wavelength ($D \ll \lambda/10$), the Rayleigh scattering is satisfied and the observed radar reflectivity is proportional to $D^6$. At commonly-used weather radar wavelengths, e.g., S- ($\sim$10 cm), C- ($\sim$5 cm), and X-band ($\sim$3 cm), the difference between radar reflectivities (in logarithmic units; dBZ) at two frequencies called dual-wavelength ratio (DWR) is usually 0 dB, since the radar wavelength is much larger than the hydrometeor dimensions. As the radar wavelength decreases to the magnitude of millimeters, e.g., K- ($\sim$8 mm) and W-($\sim$3 mm) band, which are comparable to snow dimensions, the non-Rayleigh scattering appears, and DWR > 0 dB. Therefore, the observed non-zero signatures of DWR are linked to snow sizes. For vertically-pointing radars, $DWR_{Ka,W}$ and $DWR_{X,Ka}$ start exceeding 1 dB at particle maximum dimensions of about 0.75 mm and 2 mm, respectively (Barrett et al.,



2019). Neglecting the effect of radar signal attenuation, DWR values are dependent on scattering of targets, which is the basis of dual- and triple-frequency radar retrievals of snow microphysics.

Matrosov (1993, 1998) have demonstrated that the DWR between non-Rayleigh and Rayleigh scattering frequencies can be

used to estimate particle median sizes. Scattering calculations of physically realistic snowflake shapes using discrete dipole approximation suggested that aggregation of unrimed snow leads to a "hook" feature (thick blue curve in Fig 1c) which is different from the growth of rimed snow (thick yellow curve in Fig 1c) (Kneifel et al., 2011; Leinonen et al., 2012; Leinonen and Moisseev, 2015; Leinonen and Szyrmer, 2015). Such signatures have been evidenced in airborne (Leinonen et al., 2012; Kulie et al., 2014; Chase et al., 2018) and ground-based radar observations (Kneifel et al., 2015). Furthermore, the transition

from "hook" to flat signatures in triple-frequency map is indicative of the prevailing snow growth shifting from aggregation to riming (Mason et al., 2018), which facilitates a process-level assessment of snow growth characteristics.

Understanding of snow growth obtained from ground-based triple-frequency campaigns may also bring seminal insights into next-generation satellite missions (Battaglia et al., 2020). Spaceborne dual- or triple-frequency radar retrieval algorithms have been developed based on the non-Rayleigh scattering signals among different frequency bands (Leinonen et al., 2015, 2018;

Mroz et al., 2021; Chase, 2021) and validated against in-situ observations (Mason et al., 2018; Nguyen et al., 2022). On the other hand, the small ice particles at cloud tops are good Rayleigh-scattering targets for cross-calibration among radars with different frequencies deployed on various platforms. For example, spaceborne radars, such as Tropical Rainfall Measuring Mission (TRMM, Kummerow et al., 1998), CloudSat (Stephens et al., 2002), Global Precipitation Measurement (GPM, Hou et al., 2014), Fengyun-3G (Zhang et al., 2023; Liu et al., 2024), and Earth Cloud Aerosol and Radiation Explorer (Earth-

CARE, Illingworth et al., 2015), have been proposed for cross-calibration of ground-based radars (Anagnostou et al., 2001; Li et al., 2023b). However, there is no consensus on the $Z_{Rayleigh}$ value that can be used to identify the cloud top Rayleigh scattering regions. Dolinar et al. (2022) have shown significant latitude-dependence of characteristic sizes of snow at cloud tops, suggesting the potential geographical dependence of snow sizes. In addition, cloud formation temperature and cloud type which changes with latitudes (Sassen and Wang, 2008) are relevant to snow size distributions and snow characteristic sizes

(Heymsfield et al., 2013). This impact seems to be detectable in radar observations. Hogan et al. (2000) showed that W band reflectivity starts deviating from $Z_{Rayleigh}$ ($Z_{Rayleigh,W}$) at about -20 dBZ using aircraft observations during European Cloud Radiation Experiment, while Stein et al. (2015); Li and Moisseev (2019); Matrosov et al. (2019) found it ranges from -20 to -10 dBZ in different field campaigns. Therefore, examining the geographical impact on snow microphysics also benefits the cross-calibration of multi-frequency radars.

Since 2014, a number of field campaigns hosting high-sensitivity triple-frequency radars have been carried out over regions spanning from Antarctica to tropics (Fig 1a). The well-calibrated and -aligned long-term radar observations open a new opportunity to assess snow microphysical processes over various climatologies. In this study, we attempt to compare the triple-frequency radar observations obtained in these campaigns and assess the geographical fingerprints in snow microphysical processes.





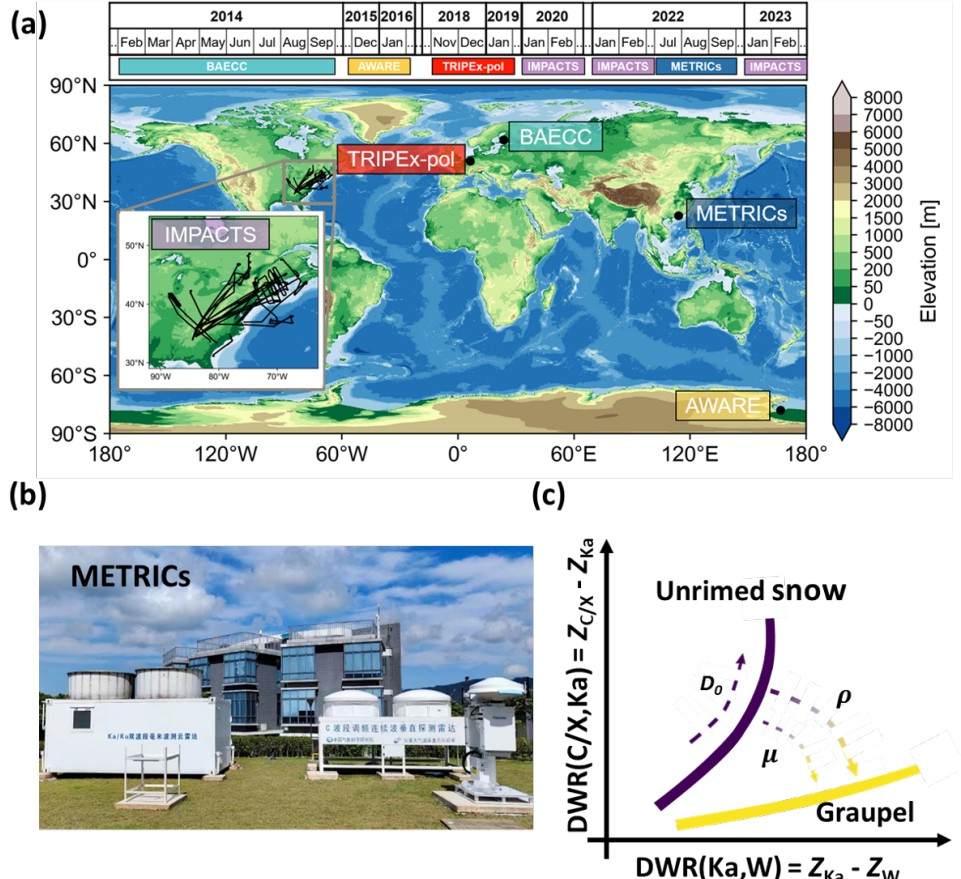

**Figure 1.** (a) Overview of the triple-frequency radar field campaigns. (b) C/Ku/Ka/W-band radars in METRICS. (c) Triple-frequency signatures of physical parameters of snowflakes. $D_0$, $\rho$, and $\mu$ denote median volume diameter, snow density, and the shape parameter of the Gamma size distribution, respectively. Since the average value of $\mu$ is around 0 (Brandes et al., 2007; Heymsfield et al., 2008; Tiira et al., 2016), our interpretation of triple-frequency statistics bears the assumption of $\mu = 0$, namely the exponential distribution. The BAECC, AWARE and IMPACTS

The remainder of this paper is organized as follows. The second section overviews five triple-frequency field campaigns, followed by the methods of snow identification, radar reflectivity calibration and riming estimation. The results are presented in the fourth section. Conclusions are given in the final section.

## 2   Data

We examine stratiform precipitation (rainfall with bright band signatures and snowfall, cloud cases were removed for minimizing the impact of sublimation) with ground-based/airborne multi-frequency radars in five field campaigns. Namely, the





Biogenic Aerosols-Effects on Clouds and Climate campaign (BAECC, Petäjä et al., 2016), TRIple frequency and Polarimetric radar Experiment for improving process observation of winter precipitation (TRIPEx-pol, Dias Neto et al., 2019; Von Terzi et al., 2022), the Atmospheric Radiation Measurements West Antarctic Radiation Experiment (AWARE, Lubin et al., 2020), Investigation of Microphysics and Precipitation for Atlantic Coast-Threatening Snowstorms (IMPACTS, McMurdie et al., 2022), and the Multi-frequency radar Experiment for TRopical Ice Clouds (METRICs). As shown in Fig 1a, these campaigns were conducted over various latitudes, ranging from tropical (METRICs), mid-latitude (IMPACTS, TRIPEx-pol), high-latitude (BAECC) to polar (AWARE) regions. Note that the X/Ka/W-band setups were employed in IMPACTS, TRIPEx-pol, BAECC and AWARE, and C/Ka/W-band radars were used in METRICs. For simplicity, we use $Z_{X/C}$ to refer X- (IMPACTS, TRIPEx-pol, BAECC, and AWARE) or C-band (METRICs) reflectivity. Similarly, $DWR_{X/C,Ka}$ denotes the dual-wavelength ratio between X- or C-band reflectivity and Ka-band reflectivity.

## 2.1 METRICs

From July to September 2022, METRICs took place at the Shenzhen Shiyan Observatory (22.65°N, 113.89°E), China. During the three-month campaign, five radars were deployed at the Shiyan observatory for synergetic observations, including vertically-pointing W-(Widener and Johnson, 2006), Ka-(Ding et al., 2022), Ku-(Ding et al., 2022), and C-band (Pang et al., 2021) profiling radars, and a L-band wind profiler (Ruan et al., 2014). The cloud radars were operating with a distance to each other less than 5 m. About 5 km to the Observatory, a X-band phased-array radar and an operational dual-polarization S-band weather radar (Li et al., 2023a) were in operation, providing reflectivity calibration basis for the C-band radar. The C-band reflectivity was calibrated by matching with the S-band radar reflectivity observations at the height of $0.5 \sim 1$ km during rainfall events. Similar with previous triple-frequency radar setups, W-, Ka-, and C-band radars were used in this study. Their range resolutions are 30 meters, and the time resolutions are 3s, 25s, 0.93s, respectively. The sonding data including temperature, pressure and humidity from Hongkong Observatory which was launched two times per day and is about 40 km from Shiyan Observatory was used for gaseous attenuation and air density correction. The stratiform rainfall observations of 20 hours were used in this study.

## 2.2 TRIPEx-pol

TRIPEx-pol was carried out at the Jülich Observatory (50.9°N, 6.4°E) (Löhnert et al., 2015; Neto et al., 2019) from November 2018 to January 2019. During TRIPEx-pol, vertically pointing X, Ka and W band radars were installed at the same roof platform with the horizontal distances less than 20 m. The X- and Ka-band systems are pulsed radar systems manufactured by Metek GmbH, while the W-band radar is a frequency modulated continuous wave (FMCW) system manufactured by Radiometer Physics GmbH. The range resolution of these radars is 30 m and the time resolutions are 2 s, 2 s and 3 s, respectively. The radar reflectivity was calibrated with the simulated reflectivity from drop size distributions (DSDs) measured by the PARSIVEL optical disdrometer during 21 rainfall periods (Von Terzi et al., 2022). Vertical profiles of atmospheric temperature, pressure, and humidity were from the European Centre for Medium-Range Weather Forecasts-Integrated Forecasting System (ECMWF-



IFS) forecasts, and we used the interpolated model products over the Jülich Observatory. In this study, the level 2 data products which have been well calibrated in total of 60-hour stratiform rainfall and 18-hour snowfall were used.

## 2.3 BAECC

The BAECC field campaign was conducted at the University of Helsinki's Hyytiälä Station (61.8°N, 24.3°E) from February to September 2014 (Petäjä et al., 2016). This experiment hosted comprehensive vertically pointing multi-frequency radars, including X/Ka-band scanning ARM cloud radar (X/Ka-SACR), Ka ARM zenith radar (KAZR) and W-band ARM cloud radar (MWACR). In this study, X-SACR, KAZR, MWAR observations were used. In addition, radiosondes were launched every six hours to obtain temperature, pressure, and humidity. The X-band reflectivity was corrected with a collocated operational C-band radar during snowfall (Falconi et al., 2018), and surface DSDs observations were used to calibrate X-band radar reflectivity at 500 m during rainfall (Li and Moisseev, 2019). In BAECC, triple-frequency radar observations of stratiform rainfall (16 h) and snowfall (12 h) were used.

## 2.4 AWARE

Within the framework of the Western Antarctic Radiation Experiment of Atmospheric Radiation Measurement (ARM), the second ARM mobile equipment (AMF2) was deployed at McMurdo Station (77.83°S, 166.67°E). The multi-frequency radars include KAZR, MWACR and X/Ka-SACR (Lubin et al., 2020), which have been used in BAECC as well. During AWARE triple-frequency radars were in operation from December 2015 to January 2016 in which the surface precipitation was in the form of snowfall. A radiosonde was launched four times a day to acquire temperature, pressure, and humidity profiles. Absolute calibrations of the scanning radar systems were performed on site with a corner reflector, and a systematic comparison was conducted with nearby CloudSat measurements (Tridon et al., 2022). The recorded snowfall was in total of 23 hours.

## 2.5 IMPACTS

IMPACTS is a multi-year field campaign with an airborne multi-frequency radar setup. Sponsored by NASA, IMPACTS was conducted to study wintertime snowstorms focusing on East Coast cyclones during the winters of 2020 ∼ 2023 (McMurdie et al., 2022). The Earth Resources 2 (ER-2) aircraft flew above snowfall systems, and carried nadir-pointing radars at frequencies of X-, Ku-, Ka-, and W-bands. The IMPACTS data has already been quality controlled including aircraft motions and attitudes (https://www.earthdata.nasa.gov/data/projects/impacts/collection), and has been employed in previous studies (Dunnavan et al., 2023; Heymsfield et al., 2023). To assure that the radar ray path is not excessively prolonged during aircraft turns, we have removed periods with rolling angles exceeding 0.2 °. We also applied the X-band reflectivity threshold of 50 dBZ to identify surface echoes. To minimize the surface clutter contamination, radar observations within 0.5 km to surface were removed. For facilitating the triple-frequency analysis, radar observations were interpolated into temporal and range resolutions of 0.5 s and 26.25 m, respectively. The hourly High-Resolution Rapid Refresh (HRRR, Blaylock et al., 2017) analysis data was





used to provide temperature, pressure and humidity for gaseous attenuation and air density correction. The quality-controlled IMPACTS dataset includes 16.5-hour snowfall and 3.5-h stratiform rainfall.

## 3 Methods

In our datasets, AWARE was focused on Antarctic snowfall, while METRICs recorded tropical rainfall. TRIPEx-pol, IMPACTS, and BAECC datasets include both rainfall and snowfall events. To generate the triple-frequency map, we identified snow in rainfall events and carefully considered different attenuation sources. To independently evaluate snow riming signatures from triple frequency observations, we quantified riming with a velocity-based approach and constructed characteristic $DWR_{X/C,Ka}$-$DWR_{Ka,W}$ curves with different riming degrees.

### 3.1 Identification of Snow

Snowfall events can be identified with surface temperature provided the absence of temperature inversion over 0 °, while it is essential to identify snow above the melting layer in stratiform rainfall events. In radar observations, the melting layer is characterized by distinct changes of polarimetric variables. The significant enhancement of linear depolarization ratio (LDR) as observed by vertically pointing radars has been widely used for melting layer detection, despite that there is slight frequency dependence (tens of meters) on the melting layer top height detection (Li and Moisseev, 2020). Here, Ka-band LDR is employed for melting layer identification. For a robust melting layer detection, we follow the principle of identifying the significant changes of LDR gradients ($d$LDR) (Dias Neto et al., 2019; Li and Moisseev, 2020; Song et al., 2021). The search for melting layer top starts from snow region, and stops at a reference level ($H_{ref}$) where the LDR is 3 dB above that in snow and $d$LDR values are consecutively positive from this level to 90 m below. To avoid potential impact of the partial melting, the melting layer top is determined 100 m above $H_{ref}$.

### 3.2 Attenuation Correction to Radar Reflectivity

Although radars in field campaigns had been well calibrated, attenuation correction needs to be considered for DWR analysis. For up-looking ground-based radars, we corrected wet radome attenuation, applied the relative calibration at the cloud top, and corrected the overestimated reflectivity due to gaseous and hydrometeor attenuation. For down-looking airborne radars in IMPACTS, we only need to account for gaseous and hydrometeor attenuation which leads to reflectivity underestimation.

Firstly, the wet radome attenuation in rainfall events was corrected by matching the observed reflectivity at 500 m and the simulated reflectivity from surface DSDs observations (BAECC, TRIPEx-pol). In METRICs, the collocated S-band weather radar (5 km away) data was used to calibrate the C-band radar reflectivity. Then, the X/C-band reflectivity profile is assumed to be well calibrated, since the attenuation from snow, rain and melting layer in stratiform rainfall is negligible (Li and Moisseev, 2019).

Secondly, the relative calibration at Ka- and W-bands was made by identifying small ice particles at the cloud top. Following the algorithm given by Tridon et al. (2020), we identified the Rayleigh-scattering regions at Ka- and W-bands using C(X)-





/Ka-band and Ka-/W-band pairs, respectively. Moving downwards from the cloud top, DWR remains constant until the non-
Rayleigh scattering at the higher frequency starts appearing. The Ka-band reflectivity was adjusted by matching the C(X)-band
reflectivity at cloud tops. Then, this approach was applied to the Ka/W-band radar pairs. The presence of strong radar signal
attenuation, e.g., in hailstorms or rainstorms, the Ka- and W-band radars suffer from sensitivity losses and their signals may
not reach the level where Rayleigh-scattering ice particles exist. Hence, the algorithm cannot detect a region of flat DWR at
cloud tops, and such profiles will be discarded.

Lastly, we need to consider the gaseous and hydrometeor attenuation. The water vapor and oxygen gaseous attenuation
was corrected using temporally interpolated sounding observations as input to millimeter wave propagation model (Liebe,
1985). The attenuation from snow and supercooled liquid water is negligible at Ka band, but can be significant at W-band.
We corrected the snow attenuation using $A_{Snow,W} = 0.0325 Z_{W,lin}$, where $Z_{W,lin}$ is $Z_W$ in linear scale (Protat et al., 2019).
This approach yielded a median W-band path-integrated snow attenuation of 0.3 dB in all field campaigns, comparing to the
value of 0.1 dB from another parameterization approach (Kneifel et al., 2015). Therefore, we can reasonably deduce that the
uncertainty of snow attenuation correction is less than 1 dB.

The attenuation from supercooled liquid water is not accounted in this study. This impact on Ka-band radar is relatively
small, but can be significant at W-band. Tridon et al. (2020) have shown that the W-band attenuation from a supercooled
liquid water path (SLWP) of 300 $gm^{-2}$ is on the magnitude of 2 dB, and Li et al. (2018) found that the occurrence of SLWP
exceeding 300 $gm^{-2}$ in snowfall is below 15 % over central Finland. From the perspective of triple-frequency signatures, the
supercooled liquid attenuation at W-band can lead to overestimated $DWR_{Ka,W}$ in IMPACTS data (top-down view), which is
similar to the effect of riming as inferred from radar Doppler velocity observations. For ground-based radars (down-top view),
the uncorrected liquid water attenuation at W-band leads to underestimation of $DWR_{Ka,W}$. Nonetheless, riming as inferred
from triple-frequency observations can be compared to those estimated from radar Doppler velocity which is immune to the
attenuation effect (Kneifel and Moisseev, 2020) as will be discussed later.

### 3.3 Doppler-velocity-based estimation of rime mass fraction

In correspondence with the triple-frequency signatures, process of riming leads to increased radar Doppler velocity (Kneifel
et al., 2015). To independently characterize the riming signatures, we use radar Doppler velocity to quantify the rime mass
fraction (Morrison and Milbrandt, 2015; Moisseev et al., 2017; Li et al., 2018),

$$FR = 1 - \frac{\int_{D_{min}}^{D_{max}} N(D) m_{ur}(D) dD}{\int_{D_{min}}^{D_{max}} N(D) m_{ob}(D) dD} \tag{1}$$

where $D_{max}$ and $D_{min}$ are maximum and minimum particle sizes, respectively; $m_{ob}(D)$ and $m_{ur}(D)$ are masses of ob-
served and unrimed snowflakes (Li et al., 2018) as a function of snow diameter $D$, respectively; $N(D)$ is the particle size
distribution. If $m_{ob}(D)$ is smaller than $m_{ur}(D)$, FR is assigned to 0. Because riming leads to increased ice density and fall
velocity, FR can also be estimated from vertically-pointing radar Doppler velocity observations (Mosimann, 1995). Following
the approach used by Kneifel and Moisseev (2020), updraft regions were removed, and a time-average window of 20 min



was imposed to cancel out up-and-downward motions. FR was estimated using the fitting between FR and the mean Doppler velocity (MDV) observed by Ka-band radars as below,

$$FR = 0.0791MDV^4 - 0.5965MDV^3 + 1.362MDV^2 - 0.5525MDV - 0.0514 \qquad (2)$$

To minimize the impact of varying air density ($\rho_{air}$), MDV was adjusted to the air condition of 1000 hPa and 0 °C (air density air, $\rho_{air,0}$) with a factor of $(\frac{\rho_{air,0}}{\rho_{air}})^{0.54}$ (Heymsfield et al., 2007). $\rho_{air}$ was derived from the temperature and relative humidity obtained from the interpolated sonde observations.

### 3.4 $DWR_{X/C,Ka}$-$DWR_{Ka,W}$ signatures of riming

In triple-frequency space, riming and aggregation can lead to diverse signatures (Leinonen and Szyrmer, 2015; Leinonen and Moisseev, 2015; Kneifel et al., 2015). To compare our observations with previous scattering simulations, we followed the approach by Leinonen and Szyrmer (2015). Firstly, we employed the backscatter cross sections of individual "realistic" dendritic snowflakes with various riming and aggregation degrees at different frequencies as simulated by Leinonen and Szyrmer (2015), and generated DWR simulations using the exponential distribution. Although the Gamma function provides a better fit for snow size distributions, the shape parameter $\mu$ is around 0 in statistics (Brandes et al., 2007; Heymsfield et al., 2008; Tiira et al., 2016), supporting the adequacy of using the exponential distribution in this study. In the exponential distribution $N(D) = N_0 exp(-\Lambda D)$, where $D$, $N_0$ and $\Lambda$ are snow diameter, the intercept parameter and the inverse scale parameter, respectively, the sizes of snowflakes are controlled by $\Lambda$. Changing $\Lambda$ from 2 $mm^{-1}$ to 367 $mm^{-1}$, we can get different characteristic sizes of snow populations and map snow growth signatures into the triple-frequency space (Leinonen and Moisseev, 2015). Then, we quantified FR using the mass-size relation from dataset as input to Equation (1). As shown in Fig. 6, the characteristic $DWR_{X/C,Ka}$-$DWR_{Ka,W}$ signatures with FR = 0, 0.53, and 0.72 are denoted by circled, squared, and star-shaped curves, respectively.

## 4 Results

### 4.1 Cloud top temperature and riming occurrence

Ice formation at the cloud top is essential for later snow growth processes. We have compared the cloud top temperature (CTT) observations in the five campaigns as shown in Figure 2. The cloud top was determined using the highest radar echo as mostly detected by the Ka band radar. In METRICs, the thick rain layer leads to decent attenuation at Ka-band and the C-band radar occasionally observed the highest cloud top (Li et al., 2025). In presence of multi-layer clouds, we have removed upper clouds with a radar echo gap exceeding 2 km for excluding the impact of seeding (Seifert et al., 2009; Li et al., 2021). As shown in Fig. 2a, clouds over tropics are characterized by the coldest tops, while cloud tops over high-latitudes are shallower and warmer, which are in line with Cloudsat observations (Sassen and Wang, 2008). Note that due to the degradation of C-band



radar sensitivity with range, the cloud top radar reflectivity in METRICs is of the magnitude of -10 ~ 0 dBZ, suggesting that the actual cloud top temperatures were even colder.

At temperatures above -70 °C, a colder cloud top is associated with more ice populations (Heymsfield et al., 2013), and the prevalence of CTT between -70 and -60 °C in METRICs is expected to generate numerous pristine ice particles. In addition, not like the stratiform precipitation in baroclinic cyclones and fronts in mid-latitudes, the stratiform rainfall over tropics is
255 closely associated with nearby convective cells (Schumacher and Houze Jr, 2003). The ice particles in stratiform region may originate from the convection at younger and more vigorous stages (Houze Jr, 1997).

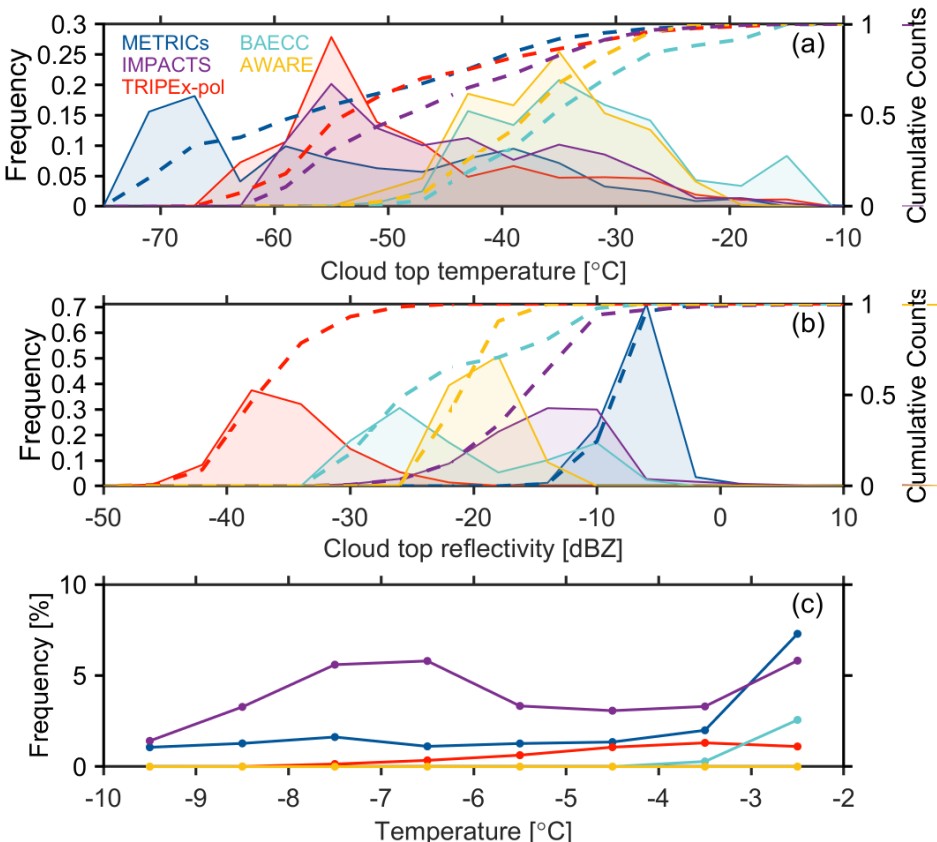

**Figure 2.** Distributions of cloud top (a) temperature and (b) corresponding calibrated radar reflectivity in the five field campaigns. The frequency of riming as defined as FR $\geq$ 0.5 as a function of in-cloud temperature is presented in (c).

Kneifel and Moisseev (2020) have shown that riming occurrence increases with temperature with seasonal variations. As shown in Fig. 2c, our statistics in general agree with their conclusions, except for rather limited riming signatures in AWARE. In addition, many rimed snow cases in IMPACTS have a surface temperature of -8 ~ -6°C and thus less observations with the
260 in-cloud temperature above -6°C, which may explain the decreased riming frequency above -6°C. Snow observations made in METRICS and IMPACTS show more frequent riming, while riming is less frequent in TRIPEx-pol and BAECC.





## 4.2 Dual-frequency signatures

### 4.2.1 Dependence on reflectivity

Statistical results of $DWR_{X/C,Ka}$ and $DWR_{Ka,W}$ as a function of $Z$ during stratiform precipitation are given in Fig. 3a,b.
As expected, DWR generally increases with $Z$ after certain thresholds below which the snow dimensions are so small that the
Rayleigh scattering condition is met. As $Z_{X/C}$ increases from 0 dBZ to about 10 dBZ, the median values of $DWR_{X/C,Ka}$
increase from 0 dB to around 1 dB (Fig. 3a). The interquartile ranges of $DWR_{X/C,Ka}$ in all campaigns except for METRICs
show great variability after $Z_{X/C}$ exceeds 10 dBZ, which is explained by the increased complexity of non-Rayleigh scattering
at Ka-band as the snow size increases. In METRICs, the interquartile ranges of $DWR_{X/C,Ka}$ remain stable near 0 dB for
$Z_{X/C}$ values below 18 dBZ. In addition, median $DWR_{X/C,Ka}$ values from both IMPACTS and METRICs do not exceed 1
dB for $Z_{X/C} < 18$ dBZ, implying the latitude dependence of DWR-$Z_{X/C}$ relation.

Similar dependence can also be found for $DWR_{Ka,W}$. As shown in Fig. 3b, $DWR_{Ka,W}$ starts exceeding 1 dB at $Z_{Ka} \approx$
0 dBZ, 2 dBZ, 4 dBZ, 8 dBZ and 12 dBZ in AWARE, BAECC, TRIPEx-pol, IMPACTS, and METRICs, respectively. We
found that $DWR_{Ka,W}$ statistics of BAECC and AWARE are similar to those observed at the Oliktok Point, Alaska (Matrosov
et al., 2019), implying similar cloud microphysics over high-latitudes and polar regions. Comparing DWR observations from
different campaigns, it appears that the radar reflectivity values over which the non-Rayleigh scattering starts appearing at Ka
(Fig. 3a) and W (Fig. 3b) bands decrease with the increase of latitude. This latitude dependence appears to be associated with
cloud top temperature (Fig. 2a). Namely, non-Rayleigh scattering signatures appear at a larger radar reflectivity threshold for a
colder cloud top.

**Table 1.** Fitted parameters for $DWR = 10^a * (Z + 20)^b$ in Figure 2.

|  | $Fitting Parameters$ | $a$ | $b$ |
|---|---|---|---|
|  | METRICs | -25.18 | 15.54 |
|  | IMPACTS | -9.334 | 5.835 |
| $DWR_{X/C,Ka}$ | TRIPEx-pol | -8.646 | 5.732 |
|  | BAECC | -7.98 | 5.346 |
|  | AWARE | -9.963 | 6.684 |
|  | METRICs | -10.44 | 6.956 |
|  | IMPACTS | -6.371 | 4.387 |
| $DWR_{Ka,W}$ | TRIPEx-pol | -5.312 | 3.826 |
|  | BAECC | -5.051 | 3.73 |
|  | AWARE | -4.18 | 3.291 |







**Figure 3.** Observed $DWR$ (circles) as a function of (a) $Z_{X/C}$ and (b) $Z_{Ka}$. The boxplots represent the median (horizontal line) and interquartile range (box) of DWR values within a reflectivity interval of 2 dB. Blue, purple, red, green and yellow dashed curves represent DWR fits for METRICs, IMPACTS, TRIPEx-pol, BAECC and AWARE, respectively. Black dot-dashed and dashed black curves represent DWR fits for snow over Oliktok Point (70.4958°N, 149.8868°W) in October 2016 and May 2017 as made by Matrosov et al. (2019), respectively. The DWR fits for expressions of $DWR = 10^a * (Z + 20)^b$ in are given in Table 1.





### 4.2.2 Dependence on temperature

Dolinar et al. (2022), using combined active and passive satellite sensors, have shown that the effective diameter of cloud top ice increases with temperature, and the ice size is generally smaller over lower latitudes. However, snow growth characteristics beneath cloud tops over different latitudes are still poorly understood. As shown in Fig. 4a,b, large $DWR_{Ka,W}$ and $DWR_{X/C,Ka}$ values ($\geq$ 3 dB) start appearing as the temperature exceeds -16 $\sim$ -12 °C, which can be explained by a pronounced difference in saturation vapor pressure between ice and liquid and a rapid aggregation favored by the dendritic features of snowflakes at around -15 °C. This implies that the unique temperature zone around -15 °C may be essential for initiating large snowflakes regardless of the regional climate.

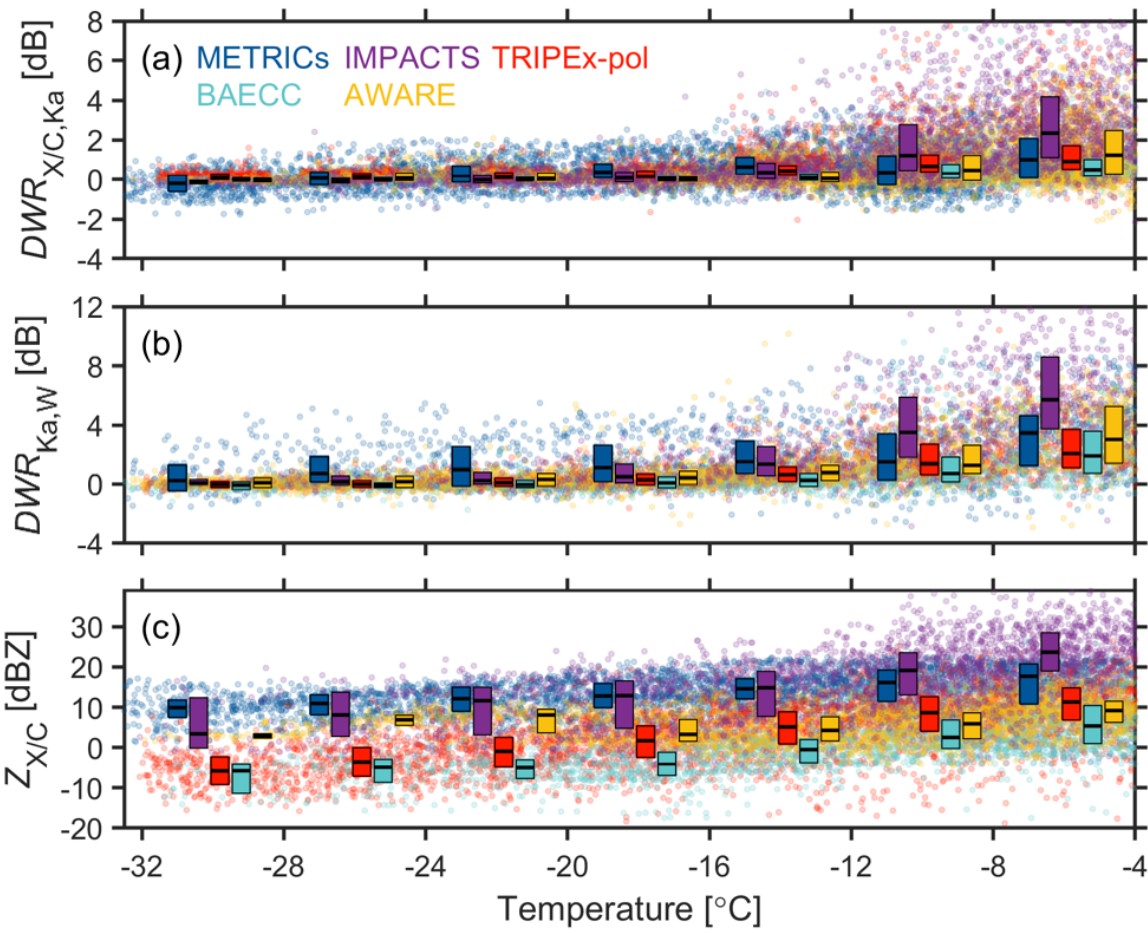

**Figure 4.** Observed (circles) (a) $DWR_{X/C,Ka}$ and (b) $DWR_{Ka,W}$ (c) $Z_{X/C}$ as a function of temperature. The boxplots represent the median (horizontal line) and interquartile range (box) of the observations within a temperature interval of 4 °C.



At temperatures above -12 °C, the rate of DWR increase with temperature is more pronounced, and $Z_{X/C}$ shows similar temperature dependence expect for AWARE. The enhanced ice growth is attributed to the thickened quasi-liquid layer on the ice particle surface, and therefore more efficient aggregation (Dias Neto et al., 2019; Li et al., 2020). In AWARE, the absence of $Z_{X/C}$ increase could be explained by the near-surface sublimation (Grazioli et al., 2017) which lowers $Z_{X/C}$ but does not necessarily lead to decrease of DWR (Tridon et al., 2022).

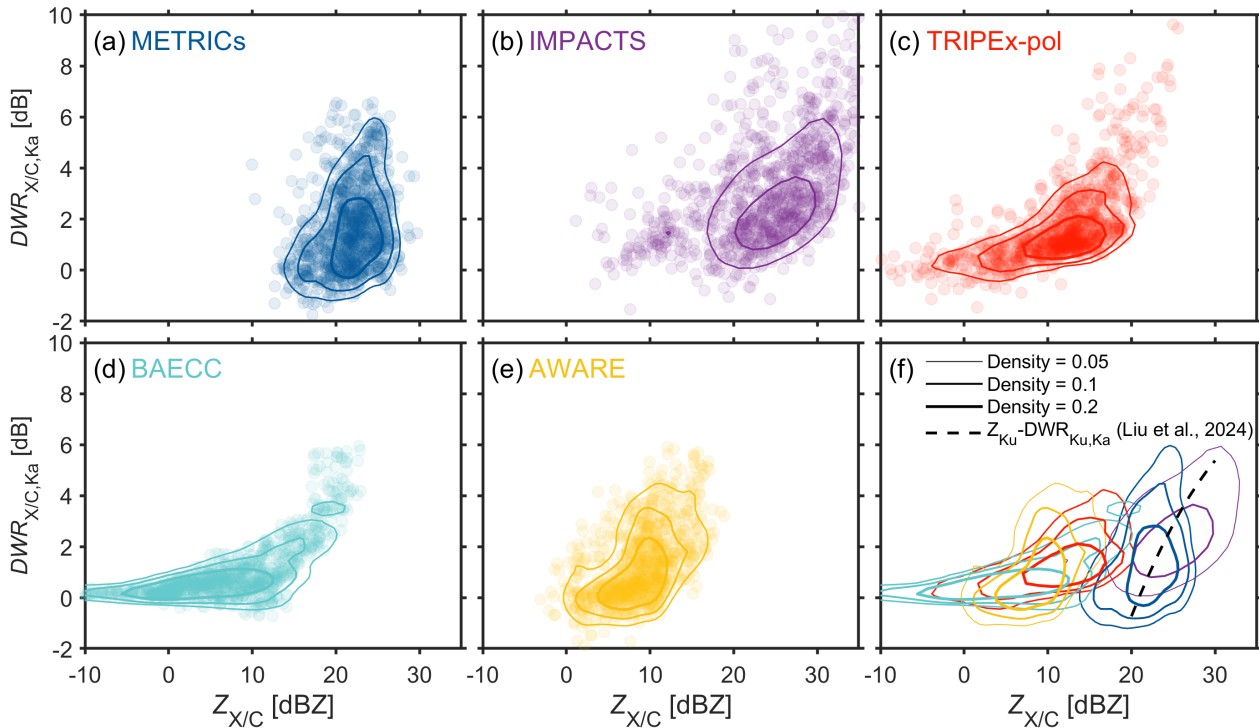

**Figure 5.** Observed $DWR_{X/C,Ka}$ (circles) as a function of $Z_{X/C}$ at temperatures greater than -8 °C in (a) METRICs, (b) IMPACTS, (c) TRIPEx-pol, (d) BAECC, and (e) AWARE. The colored isolines represent the observation density distributions for each field campaign. In (f), the black dashed curve marks the fit for FY-3G Ku and Ka band radar observations of snow over the melting layer of low- to mid-latitude stratiform rainfall (Liu et al., 2024).

A zoom-in view into the $Z_{X/C}$-$DWR_{X/C,Ka}$ space for temperatures above -8 °C (Fig. 5) reveals an obvious positive correlation between $DWR_{X/C,Ka}$ and $Z_{X/C}$. For a given $Z_{X/C}$ at 20 dBZ, $DWR_{X/C,Ka}$ in METRICs and IMPACTS is mostly $0 \sim 2$ dB, comparing to $2 \sim 4$ dB in TRIPEx-pol and BAECC, suggesting that a high $Z_{X/C}$ does not necessarily mean large-sized snowflakes. Snow observations with $Z_{X/C} \geq 20$ dBZ are prevalent in METRICs and IMPACTS, but they are rather rare in TRIPEx-pol, BAECC and AWARE. Recently, Liu et al. (2024) using FY-3G spaceborne radar observations of snowflakes above melting layers of low- to mid-latitude stratiform rainfall showed that $DWR_{Ku,Ka}$ stably increases with $Z_{Ku}$, and their fit (black dashed curve in Fig. 4) actually agrees well with the ground-based radar observations made in METRICs

 

and IMPACTS. Given cloud top temperatures in METRICs and IMPACTS are lower than -70 ∼ -50 °C (Fig. 2a), the fit in (Liu et al., 2024) appears to be representative of low- to mid-latitude deep precipitating snowfall events.

## 4.3 Triple-frequency signatures

Observed $DWR_{X/C,Ka}$-$DWR_{Ka,W}$ occurrence and FR for temperature ranges of -10 ∼ 0 °C are given in Fig. 6. As the temperature rises from -20 ∼ -10 °C (gray isolines) to -10 ∼ 0 ° (white isolines), the increases of $DWR_{X/C,Ka}$ are about 2
∼ 6 dB in METRICs, IMPACTS, TRIPEx-Pol and BAECC. In contrast, the weak snow growth in AWARE may be because of the katabatic winds which lower relative humidity (Grazioli et al., 2017) and the sparse amount of INPs over McMurdo (Hines et al., 2021).

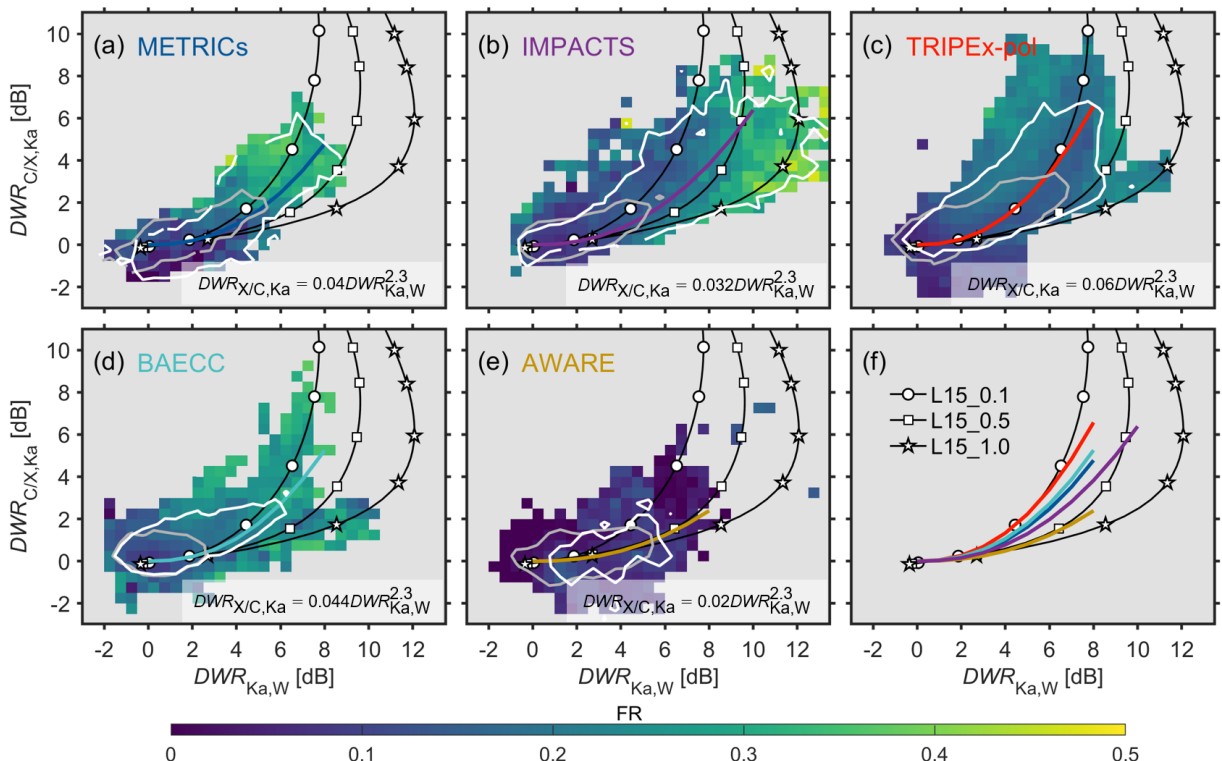

**Figure 6.** Observed $DWR_{X/C,Ka}$-$DWR_{Ka,W}$ occurrence and FR within the temperature range of -10 ∼ 0 °C from (a) METRICs, (b) TRIPEx-pol, (c) BAECC, and (d) AWARE. The DWR planes are overlapped by scattering model curves of rimed dendrite aggregates with effective liquid water paths of 0.1 kg m$^{-2}$ (FR ≈ 0) , 0.5 kg m$^{-2}$ (FR ≈ 0.53), and 1 kg m$^{-2}$ (FR ≈ 0.72) (L15_0.1, L15_0.5, and L15_1.0, respectively) as adapted from (Leinonen and Szyrmer, 2015). The gray and white isolines represent the $DWR_{X/C,Ka}$-$DWR_{Ka,W}$ occurrence density of 5% within the temperature of -20 ∼ -10 °C and -10 ∼ 0 °C, respectively. The fitting curves are marked with different colors.




In METRICs, IMPACTS and TRIPEx-pol, 5% of $DWR_{X/C,Ka}$ observations between -10 °C and 0 °C (white isolines) extend to above $5 \sim 6$ dB. Specifically, triple-frequency observations in IMPACTS generally follow the scattering model assuming the FR of $0.53 \sim 0.72$ and the velocity-based FR estimates are on the order of $0.4 \sim 0.5$, presenting obvious riming signatures. The observed $DWR_{X/C,Ka}$-$DWR_{Ka,W}$ signatures in METRICs and IMPACTS follow the scattering model assuming the FR = 0, while the velocity-based FR in METRICs is larger than TRIPEx-pol. Since the uncorrected supercooled liquid water attenuation does not affect the observed Doppler velocity but can lead to underestimated $DWR_{Ka,W}$, It is anticipated that the observed $DWR_{Ka,W}$ in METRICs is underestimated. The highest $DWR_{X/C,Ka}$ observations and least riming signatures were found for TRIPEx-pol, implying that the absence of riming is favorable for snow aggregation (Li et al., 2020; Chellini et al., 2022). To compare the triple-frequency signatures in these campaigns, power-law fits were made to the median values of $DWR_{X/C,Ka}$ in each $DWR_{X/C,Ka}$ interval. As shown in Fig. 6, the prefactor in AWARE is the smallest, and is the largest in TRIPEx-pol.

## 5   Conclusion and Discussion

In this work, we examined snow microphysics over various geographies using triple-frequency radar observations from MET-RICs, TRIPEx-pol, BAECC, IMPACTS and AWARE. Our results suggest the promise of using long-term triple-frequency setup for understanding the climatology of ice formation, snow aggregation and riming processes. Our major conclusions are conceptualized in Fig. 7 and summarized below,

1. Majority of snow in BAECC and AWARE (high latitudes) originates from a cloud top temperature > -40°C, and therefore the heterogeneous nucleation plays an essential role in snow formation. In contrast, precipitating clouds in other campaigns (low- to mid-latitudes) are much deeper, and the coldest cloud top was found in METRICs.

2. Following the conceptual model given by Kneifel et al. (2015), our analysis suggests that the triple-frequency signatures of riming generally collaborate with the velocity-based approach, and major snow growth characteristics are dependent on regional climate. With the in-cloud Doppler measurement capability, long-term EarthCARE observations may be used for globally mapping riming signatures. The velocity-based FR estimates suggests increased frequency of snow riming with in-cloud temperature. The heaviest riming in IMPACTS, corroborating with the expected riming signatures in $DWR_{X/C,Ka}$-$DWR_{Ka,W}$ space. In contrast, riming signatures in TRIPEx-pol and AWARE is rather weak. Although the multi-month triple-frequency observations in previous field campaigns are not adequate for the analysis of seasonal variations and aerosol-snow interactions, long-term triple-frequency observations are expected to shed novel insights snow microphysics climatology in the future.

3. DWR observations in these field campaigns qualitatively indicate the dendritic growth zone around -15 ° playing a key role in initiating enhanced snow size growth, and reveals a generally temperature-dependent snowflake growth characteristics. Specifically, TRIPEx-pol presents a typical "hook" signature with FR = 0, indicative of favorable aggregation in absence of riming. The weakest snow growth is found for AWARE, potentially owing to the scarcity of ice nucleating particles and avaliable water vapor for deposition.





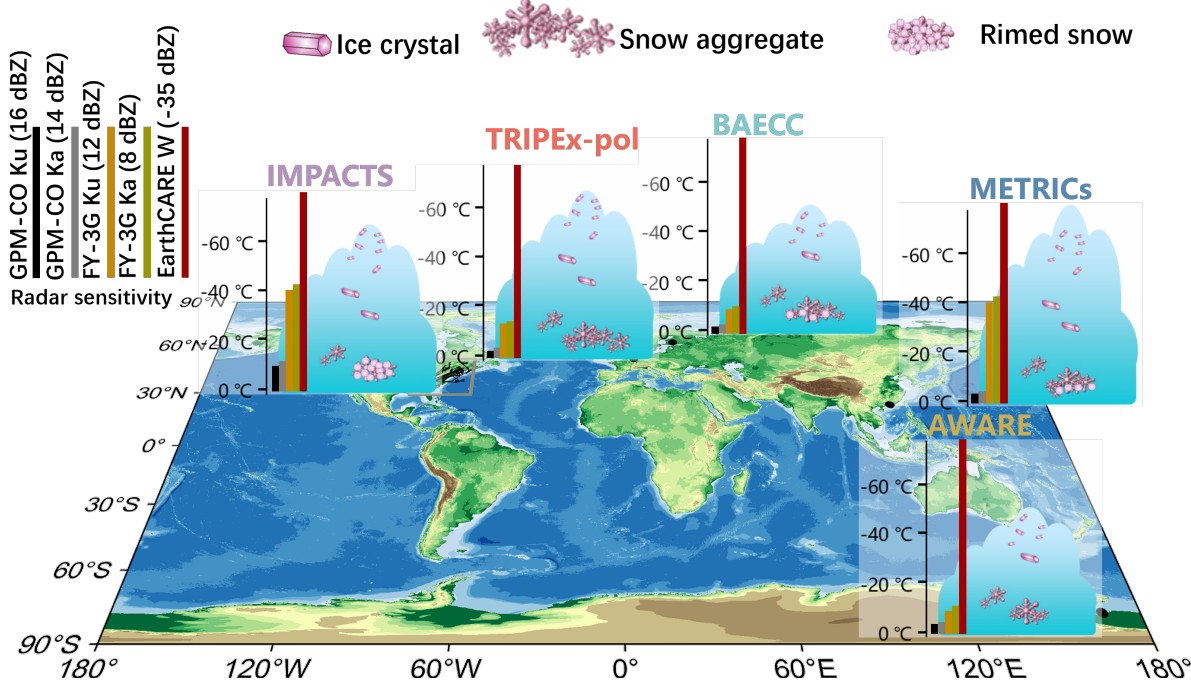

**Figure 7.** Conceptual diagram of major conclusions in this study. Based on the statistics in Fig.4c and current spaceborne radar sensitivities, we sketched the detectable temperature ranges above the melting layer over different regions with colored bars.

4. Our statistics are also indicative of the capability of current spaceborne radars in snow detection. Compared to the high sensitivity of EarthCARE W-band radar, the sensitivity of GPM-CO and FY-3G radars is on the magnitude of 10 - 20 dBZ. Namely, GPM-CO and FY-3G radars well detect low- to mid-latitude snow at temperatures above -12 °C but most high-latitude snow cannot be observed (Fig. 4).

On the other hand, the minimum reflectivity where the Ka-band non-Rayleigh scattering appears over low- to mid-latitudes (METRICs and IMPACTS) is about $15 \sim 20$ dBZ ($Z_{X/C}$) which is well detectable by FY-3G Ku-(12 dBZ sensitivity) and Ka-band (8 dBZ sensitivity) radars (Liu et al., 2025). This suggests that snow above the melting layer if is well identified could be served as Rayleigh-scattering targets at both Ku and Ka bands for inter-calibration of ground-based S-band weather radars and spaceborne radars without considering the attenuation effects.

In the last decade, ground-based/airborne triple-frequency radar field campaigns have demonstrated the unprecedented value of triple-frequency radars in unraveling snow microphysics in various geographics. The recent success of METRICs campaign allowed us to do synergetic analysis of the snow growth climatology from polar regions to tropics, and significant geographical differences have been identified. Thanks to the high-sensitivity of well-matched ground-based/airborne triple-frequency radars, the snow growth processes with rather weak radar echoes are detectable. In contrast, spaceborne radars can also provide very

unique W- (CloudSat, EarthCARE), Ka- (GPM, FY-3G) and Ku-band (GPM, FY-3G) coincidence observations (Turk et al.,



2021), while the minimal detectable signal is on the order of 13 ~ 19 dBZ for GPM Ku/Ka-band radars (Masaki et al., 2020), and 10 ~ 13 dBZ for FY-3G Ku/Ka-band radars (Liu et al., 2024, 2025). Namely, the majority of snow growth signatures (Fig.3, Fig.4) as observed in the five campaigns are missed in these dataset. For the future Ku-Ka-W band spaceborne mission, Leinonen et al. (2015) assumed a minimal sensitivity of 0 ~ 5 dBZ which seems to be a good selection for making use of the

non-Rayleigh scattering characteristics (Fig. 3) and majority of solid precipitation above -10 °C can be detected (Fig. 4c).

*Data availability.* The used data including METRICs in this paper is available at https://zenodo.org/records/14575727. TRIPEx-pol observations can be accessed at https://zenodo.org/record/5025636. Observations made in BAECC and AWARE are available from ARM data center (https://www.arm.gov/data/). IMPACTS observations can be accessed at https://www.earthdata.nasa.gov/data/projects/impacts/collection.

*Author contributions.* HL and XS conceptualized the study. QL and HL performed the experiment and wrote the paper. YZ, WL, ZR, LL,
AL, and CZ conducted the METRICs campaign and collected the radar observations. All the authors took part in the interpretation of the results and edits of the paper.

*Competing interests.* The authors declare that they have no conflict of interest.

*Acknowledgements.* We thank Dr Stefan Kneifel, Dr Leonie Von Terzi, and Dr Maximilian Maahn for very helpful comments on this work. This research has been supported by National Natural Science Foundation of China (42475095), Open Grants of the High Impact Weather
Key Laboratory (special), China Meteorological Administration (2024-K-01), Basic Research Fund of CAMS (2023Z008), Anhui Provincial Natural Science Foundation (2408055UQ007) and Open Grants of the China Meteorological Administration Aerosol-Cloud and Precipitation Key Laboratory (KDW2412), and Alexander von Humboldt Foundation.



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
