# Peer review of "A survey of snow growth signatures from tropics to Antarctica using triple-frequency radar observations"

_EGUsphere, 2025_

## Author Comment (AC1)

**Reviewer #1:**

This study investigates snow growth characteristics using triple-frequency radar observations from multiple field campaigns. Based on early seminal concept connecting snow microphysics to triple-frequency signatures, this study demonstrates how these early findings can be used to compare the climatology of snow microphysics and benefit future triple-frequency satellite missions. In particular, the measurements at Southern China are unique, and I congratulate the authors for generating an important dataset. The manuscript is well written, organized, and clear. I have nothing substantive to add, aside from some minor suggestions listed below.

We sincerely thank the reviewer for providing constructive comments and suggestions on this manuscript. Your positive feedback is greatly appreciated and fuels our commitment to advancing triple-frequency radars. Please see below our detailed point-by-point responses:

Comments:

(1) L66: The Introduction is a good summary of current understanding of triple frequency technology and its applications. I suggest adding some discussion on how such observation may benefit model development.

Thank you for your suggestion. Although most studies are focused on the triple frequency retrievals, model applications should be definitely mentioned. We have added the discussions below:

*Tridon et al (2022) identified unusual triple-frequency signatures of riming over Antarctica, and discussed responsible factors using model simulations. Karrer et al. (2021) using triple-frequency observations validated model parameterizations of snow aggregation, which confirmed the need to represent an additional peak in snow sticking efficiency at -15°C. In addition, forward simulations of spaceborne triple-frequency observations indicate that at least one frequency can be reliably employed to monitor the entire cloud-precipitation process (Leinonen et al., 2015; Wang et al., 2019).*

(2) L198: Why a single A-Z parameterization is used?

Thank you for your question. The attenuation from snow and supercooled liquid water is negligible at Ka band, but can be significant at W-band. We did give thinking on the applicability of using a single A-Z relationship. To address this question, we have compared two parameterization approaches for W-band path-integrated snow attenuation estimation as given in (Protat et al. 2019) and (Kneifel et al., 2015), respectively. The median values from the five

campaigns are about 0.3 dB and 0.1 dB, respectively (see details in the figure below). Therefore, we expect that the uncertainty of using a single fit (Protat et al. 2019) is less than 1 dB.

[Figure]

Path-integrated Attenuation at W band in each field campaign. Wcorr1 and Wcorr2 denote methods used in (Protat et al. 2019) and (Kneifel et al., 2015), respectively

(3) L279: A more detailed physical interpretation of the variations of DWR-Z dependence, as well as how cloud-top temperature affects the radar threshold for non-Rayleigh scattering, is needed.

This is a valid concern. We have added the following discussions on DWR-Z dependence.

*The average cloud top over low-latitudes is above 15 km, but is below 6 km at high-latitudes (Bertrand et al., 2024). Namely, the climatology of cloud top temperature increases towards polar regions. Given much colder cloud tops corresponding to more INPs and the ice transport from local convections over low-latitudes, ice number concentrations are expected to decrease with latitude. In contrast, significant increase of DWR(Ka,W) occurs below the dendritic growth zone across all campaigns (Fig. 4b), indicating that in-cloud temperature is more important than ice number concentration for snow aggregation. Hence, larger $Z_{X/C}(Z_{Ka})$ thresholds for Ka- (W-) band non-Rayleigh scattering at METRICs and IMPACTS may be explained by more ice particles over low latitudes.*

(4) Fig. 6: The comparison is important. It demonstrates that the velocity-based FR estimate is a good approximate for heavy riming. Does this mean that we do not need W-band radar considering the attenuation effect, if the velocity is accurately measured?

This is a valid concern. We have added the following discussions in Section 4.3,

*Although the velocity-based FR is qualitatively consistent with triple-frequency signatures of riming, FR should be used with caution. Kneifel & Moisseev (2020) have shown that large uncertainties exist for FR < 0.5, which is actually the reason why we quantified riming occurrence using FR >0.5 in Fig. 2c. We had proposed a dual-frequency approach for riming classification (Li et al., 2020), while the triple-frequency observations are more favorable for quantitative retrieval of snow microphysics (Mason et al., 2019). In addition, the use of third frequency (X- or C-band) narrows the retrieval of snow size distributions. Although the exponential function for snow size distribution can be assumed in climatological analysis, the third frequency is needed to inform the parameters in Gamma function which is more consistent to observations (Mason et al., 2019).*

(5) L320: Ideally, much more data should be used for a climatological analysis. So, the impact of relatively short observation period for current results should be discussed.

We agree with this comment. We have added a paragraph in Conclusions for discussing this point.

*While the promise of using multi-month triple-frequency radar observations is presented in this study, the analyzed field campaigns were conducted in different seasons. In addition, the datasets generated from relatively short observation periods are not adequate for examining seasonal variations. We advocate the sustained support for a triple-frequency super site, which would facilitate multi-year observations and greatly advance our climatological understanding of snow microphysics.*

---

## Author Comment (AC2)

**Reviewer #2:**

This study investigated triple frequency radar data from five locations and discussed ice microphysics for snow riming in those locations. The results seem reasonable and to be interpret reasonably. Figures are clear, and the text is well organized. Some of the datasets (e.g. AWARE, TRIPEx-pol, METRICs) might not be enough to explain climatology or general characteristics of snow at the site, but this study got foot in door of analyzing climatological/statistical characteristics of snow riming using the multi-frequency radar datasets. The manuscript should be published after addressing minor comments below.

We sincerely thank the reviewer for providing constructive comments and suggestions on this manuscript. We acknowledge the insufficient observation period for a solid climatological analysis, and thank you for recognizing our contributions. Please see below our detailed point-by-point responses:

Comments:

(1) Please add an advantage of using triple frequency rather measurements than dual frequency measurements or Dual frequency + Doppler measurements. The discussions of snow riming and aggregation are well explained by the triple frequency data analysis, but I felt that the discussions in the manuscript can also be explained using the dual frequency result only (DWR Ka,W and Zka + FR). Please more highlight the advantage of adding the third frequency (i.e. X or C) and what could be clarified by the triple frequency data analysis and not be clarified by the dual frequency (+FR) analysis only.

Thank you for your suggestion. We have added the following discussions in Section 4.3,

*Although the velocity-based FR is qualitatively consistent with triple-frequency signatures of riming, FR should be used with caution. Kneifel & Moisseev (2020) have shown that large uncertainties exist for FR < 0.5, which is actually the reason why we quantified riming occurrence using FR >0.5 in Fig. 2c. We had proposed a dual-frequency approach for riming classification (Li et al., 2020), while the triple-frequency observations are more favorable for quantitative retrieval of snow microphysics (Mason et al., 2019). In addition, the use of third frequency (X- or C-band) narrows the retrieval of snow size distributions. Although the exponential function for snow size distribution can be assumed in climatological analysis, the third frequency is needed to inform the parameters in Gamma function which is more consistent to observations (Mason et al., 2019).*

(2) Could you explain more why non-Rayleigh scattering signatures appear at cases when large radar reflectivity at a colder cloud top? I am interesting in discussion of microphysics of this characteristics accounting for the environments at the site.

This is a valid concern. Please see our response to your last question.

(3) Lines 285-286: I cannot see this signature for the METRICs and BAECC datasets.

After carefully checking our observations, we found that this signature is not that obvious in DWR$_{X/C,Ka}$. We have amended this sentence as below,

*As shown in Fig. 4a, large DWR$_{X/C,Ka}$ values ($\geqslant$ 3 dB) are concentrated at temperatures exceeding -12 ℃. More significant temperature influence is found for DWR$_{Ka,W}$ (Fig. 4b), which can be explained by a pronounced difference in saturation vapor pressure between ice and liquid and a rapid aggregation favored by the dendritic features of snowflakes at around -15 ℃.*

(4) I would prefer stating "warmer/colder" than a specific temperature rather than using "above/below" a specific temperature. Because I assumed that the colder temperature represents higher altitude, I was sometime confused with this about the altitudes.

Corrected.

(5) I am interested in the following discussions, and it would be great to add the discussions in the manuscript.

How/Why the warmer cloud top temperature correlates with high DWR?

Why high latitude sites have warmer cloud top and heterogeneous nucleation? Considering that high latitude sites could have lower aerosol concentrations, heterogeneous nucleation could be less. Please give some comments about this.

Thank you for your questions.

Q1. It is intriguing to see this correlation in Fig.3. Actually, this is related to ice number concentration. As shown in Fig. 4b, the increase of DWR is largely dependent on temperature. Since more ice particles are expected over low latitudes, their reflectivity is larger when ice particles are relatively small (T < -15℃). When we see a significant increase of DWR, the reflectivity at low latitude is therefore higher.

Q2. The warmer cloud top at high latitudes is associated with cloud vertical structure. As shown by Bertrand et al (2024), the climatology of cloud top height decreases with latitude, and the cloud top temperature over high-latitudes is generally warmer than low latitudes (Fig. 2a).

[Figure]

Cloud fraction as a function of latitude (Bertrand et al., 2024)

To better clarify this point in the manuscript, we have added some discussions as below,

*The average cloud top over low-latitudes is above 15 km, but is below 6 km at high-latitudes (Bertrand et al., 2024). Namely, the climatology of cloud top temperature increases towards polar regions. Given much colder cloud tops corresponding to more INPs and the ice transport from local convections over low-latitudes, ice number concentrations are expected to decrease with latitude. In contrast, significant increase of DWR(Ka,W) occurs below the dendritic growth zone across all campaigns (Fig. 4b), indicating that in-cloud temperature is more important than ice number concentration for snow aggregation. Hence, larger $Z_{X/C}(Z_{Ka})$ thresholds for Ka- (W-) band non-Rayleigh scattering at METRICs and IMPACTS may be explained by more ice particles over low latitudes.*